# The Effect of Long-Term Use of an Eyewash Solution on the Ocular Surface Mucin Layer

**DOI:** 10.3390/ijms20205078

**Published:** 2019-10-13

**Authors:** Hiroyuki Yazu, Naoyuki Kozuki, Murat Dogru, Ayako Shibasaki, Hiroshi Fujishima

**Affiliations:** 1Department of Ophthalmology, Keio University School of Medicine, Shinjuku-ku 160-8582, Tokyo, Japan; muratodooru2005@yahoo.co.jp; 2Department of Ophthalmology, Tsurumi University School of Dental Medicine, Tsurumi-ku 230-8501, Kanagawa, Japan; naoskds@gmail.com (N.K.); bce00432@yahoo.co.jp (A.S.); fujishima117@gmail.com (H.F.); 3Kozuki Eye Clinic, Setagaya-ku 156-0052, Tokyo, Japan

**Keywords:** eyewash, mucin, MUC5AC, MUC16

## Abstract

The use of eyewash solutions in Japan, especially in patients with allergic conjunctivitis and contact lens wearers, has been increasing. Our aim was to investigate how the use of preservative-free eyewash solution in healthy eyes for one month affects corneal safety and ocular surface mucin. We analyzed 42 eyes of 21 individuals (17 males, four females; mean age: 36.1 ± 7.4 years) without ocular allergies, dry eyes, or other ocular diseases through a prospective study. Eyes were randomized to a wash group (group one) and a nonwash follow up group (group two). We evaluated the dry eye-related quality-of-life score (DEQS), tear film breakup time (TBUT), fluorescein staining score, mRNA expression of MUC5AC and MUC16, MUC16 immunohistochemistry, and MUC5AC periodic acid Schiff (PAS) staining. There was a significant decrease in DEQS scores after one month of eyewash use (*p* < 0.05). There were no significant differences in other evaluation items that were analyzed (all *p* > 0.05). Furthermore, no significant differences were observed between group one and group two in all endpoints (all *p* > 0.05). The results suggest that one month use of a nonpreserved eyewash solution has no detrimental effects on the tear film and the ocular surface mucins.

## 1. Introduction

The use of eyewash solutions in Japan, especially in patients with allergic conjunctivitis and contact lens wearers, has been increasing. A cup type eye wash solution, Eyebon^®^ (Kobayashi Pharmaceutical Co., Ltd.; Osaka, Japan) was launched in 1995 triggering wide recognition of eyewash solutions. In the past, eyewash solutions contained preservatives (benzalkonium chloride [BAK]) until 2002. Corneal epithelial disorders associated with the use of eyewash solutions have been reported [1]. Currently, preservative-free eyewash solutions have become popular. We have already reported the ocular safety of nonpreserved eyewash solutions for allergic conjunctivitis [2,3]. Moreover, problems caused by commercial eyewash, for example, infection and dry eyes, have not been reported yet, however, many physicians are concerned that washing eyes may alter the structure of the mucin in the tear film and may promote infection and epithelial keratoconjunctivitis.

According to the Japanese guidelines for allergic conjunctival diseases [4], the use of eyewash solutions is not recommended due to the risk of skin contamination around the eyes and introduction of antigens from the skin to the ocular surface, however, these claims have not been proven. The effects of long-term eyewash use on the ocular surface have not been studied so far.

Ocular surface mucins are believed to provide a barrier to prevent pathogens and particulate matter from entering the ocular surface epithelium and their heavy O-linked glycosylation maintains hydration of the ocular surface and tear stability [5]. It has been reported that four subtypes of secreted mucins (MUC2, MUC5AC, MUC7, and MUC19) and four types of the membrane-associated mucins (MUC1, MUC4, MUC16, and MUC20) are expressed on the ocular surface [6]. MUC5AC is a gel-forming mucin secreted from the conjunctival goblet cells and it is expressed mostly on the ocular surface. The expression of MUC16 in the ocular surface glycocalyx helps provide a nonadhesive protective barrier for the epithelial surface [7]. We reported that the MUC5AC expression levels decreased significantly in the conjunctival epithelial cells in patients with atopic keratoconjunctivitis, a severe allergic conjunctival disease, whereas MUC16 expression was upregulated [8], however, the effects of eyewash on mucin and tear stability have not been fully understood and elucidated. 

In this study, we investigated the effect on corneal safety and ocular surface mucin (MUC5AC and MUC16) after use of a preservative-free eyewash solution in healthy eyes for one month.

## 2. Results

### 2.1. Changes in the dry eye-related quality-of-life score (DEQS) Scores with Eyewash Use

The mean DEQS score at baseline was 16.0 ± 12.9, and at one month after eyewash treatment it was 4.6 ± 5.7 in group one and 7.9 ± 10.6 in group two. There was a significant decrease after one month of eyewash use (*p* = 0.001), whereas in group two, there was no significant difference between baseline and one month later. (*p* > 0.05) (Figure 1).

### 2.2. Changes in the TBUT and Ocular Surface Staining Scores

Figure 2 shows the changes in the TBUT one month after eyewash use. The mean TBUT at baseline showed no significant difference among the groups (group one: 8.07 ± 3.08 s, group two: 7.69 ± 2.14 s; *p* > 0.05). Similarly, there was no significant difference at one month after initiation of eyewash treatment (group one: 7.76 ± 2.15 s, group two: 7.66 ± 2.30 s; *p* > 0.05). In the comparisons within the subgroups, there were no significant deteriorations after one month (both group one and group two; *p* > 0.05).

None of the objects had any corneal and conjunctival staining before and after one month of eyewash use. 

### 2.3. Changes in the MUC5AC and MUC16 mRNA Expression Levels

Figure 3 shows the MUC5AC and MUC16 mRNA expression levels. For MUC5AC, there were no significant decrease in number of copies/ng RNA (GAPDH) from baseline (group one: 0.02 ± 0.03, group two: 0.03 ± 0.05) to one month after initiation of eyewash use (group one: 0.02 ± 0.03 group two: 0.03 ± 0.03) (*p* > 0.05). Moreover, no significant difference in MUC5AC mRNA expression between washed and unwashed eyes after one month was observed (*p* > 0.05). For MUC16, there was no significant decrease in number of copies/ng RNA (GAPDH) from baseline (group one: 0.29 ± 0.10, group two: 0.31 ± 0.18) to one month after initiation of eyewash use (group one: 0.33 ± 0.19 group two: 0.33 ± 0.24) (*p* > 0.05). In addition, no significant difference in MUC16 mRNA expression between washed and unwashed eyes after one month was observed (*p* > 0.05).

### 2.4. Changes in the MUC16 Immunohistochemistry Staining

Figure 4 shows the representative MUC16 immunohistochemistry staining before and after one month of eyewash use in each group. Immunohistochemical staining of the conjunctival imprints from eyes showed similar positive staining for MUC16. The ratios of the areas stained with MUC16 antibodies to the overall areas of conjunctival epithelial cells did not show significant differences at the baseline (group one: 1.5 ± 1.3%, group two: 2.9 ± 3.3%) and one month after eyewash use (group one: 2.0 ± 1.9 %, group two: 2.3 ± 1.7%) (*p* > 0.05)(Figure 5). 

### 2.5. Changes of MUC5AC PAS Staining and Goblet Cell Density

Figure 6 shows the representative MUC5AC periodic acid schiff (PAS) staining before and after one month of eyewash use in each group. PAS staining of the conjunctival imprints from eyes showed positive staining for numerous MUC5AC + goblet cells and areas of mucin pick up with sheets of healthy conjunctival epithelial cells. Figure 7 shows the comparisons of conjunctival goblet cell density between group one and group two at baseline (group one: 14.1 ± 23.6 cells/hpf, group two: 18.0 ± 18.7 cells/hpf) and one month after eyewash (group one: 15.3 ± 15.0 cells/hpf, group two: 18.5 ± 12.6 cells/hpf). There were no significant differences in relation to both within-subgroup and between-subgroup variations (all *p* > 0.05). 

## 3. Discussion

In this study, we found that a preservative-free eyewash solution had no adverse effects on ocular surface mucins, MUC5AC, and MUC16 and did not induce epithelial damage. We have previously reported that eyewash is effective for the removal of allergies and foreign bodies [2]. Furthermore, this study shows that eyewash does not significantly alter the stability of tears on the ocular surface. Iwashita et al. reported that eyes with Eyebon^®^ wash had significantly lower fluorescein and lissamine green staining scores as compared with the physiological saline solution containing BAK [3]. Their results indicated that BAK could aggravate eye diseases such as dry eyes and superficial punctate keratitis (SPK). Therefore, research on mucin secretion and mucin-secreting cells can provide invaluable information and new perspectives for the management of ocular surface disorders including dry eyes. Alteration of membrane-associated mucin expressions and gel-forming mucin secretion by goblet cells have been noted in ocular surface diseases including dry eyes [9]. 

Dry eye disease is a common and a major reason for visits to eye clinics. It is reported that 7.4–33.4% of the worldwide population has been diagnosed with dry eye (560 million to 2.54 billion patients) [10,11] Dry eye is characterized by the loss of tear volume, rapid breakup of the tear film, and evaporation of tears. It is proposed that TBUT is the one of the core objective findings in dry eye diagnosis, and it induces declines in visual performance and optical quality [11]. Because of the increased incidence of this disease in Asia, the diagnostic criteria of dry eye were renewed by the Asia Dry Eye Society [12]. The renewed criteria highlight an essential role of TBUT assessment and defined TBUT as the most important objective phenotype in dry eye patients. Although abnormality of mucin is one of the causes of decreased TBUT (unstable tear film) and is common, it is rarely evaluated routinely in clinical practice setting [9]. 

In this study, the patients were relatively young and dry eye patients were excluded. According to Farrand et al., the prevalence of dry eye increased with age and was higher among women than men [13]. Therefore, age and sex-related differences in relation to ocular surface changes with eye wash should be elucidated in the future. Eyebon^®^ contains chondroitin sulfate sodium, which has a protective effect on the cornea, and dipotassium glycyrrhizinate, which has an anti-inflammatory effect. Chondroitin sulfate sodium has also been shown to be effective in the treatment of corneal epithelial defects and ulcerative keratitis in animal models [14]. Shimazaki-Den et al. reported that MUC5AC expression levels in the conjunctival epithelium were significantly lower in patients with dry eye [15]. Argueso et al. reported that MUC5AC expression was significantly lower in the tear fluid and conjunctival epithelial cells in patients with Sjögren syndrome as compared with normal individuals [16]. The effects of using eyewash solutions in eyes with such diseases and low mucin expression levels should be investigated soon. This might provide interesting information. If the microenvironment of the ocular surface in such patients is different from that in healthy people, it cannot be denied that mucin is decreased. Therefore, in the present study, we intended to study only healthy people and then patients with any ocular diseases in the future, however, there are ethical considerations in Japan, and therefore it is a future issue. Eyebon^®^ is added as the pH regulator because it could be adjusted to the same pH as the tear to prevent a burning sensation. We cannot deny the possibility that excipients such as polysorbate 80, l-menthol, and dl-camphor also may have activity of epithelial damage such as BAK, but we did not evaluate it in this study. Thus, further study to examine whether corneal epithelial disorders can be caused by excipients is needed.

This study showed no significant changes in the objective evaluations (TBUT and ocular surface staining) before and after washing the eyes, whereas the subjective evaluation (DEQS) had significantly improved after eyewash. Because dry eye is diagnosed by a combination of subjective symptoms and decreased TBUT [12] in Japan, our findings suggested that the possibility of exacerbating dry eye conditions by eyewash may be very low, however, if the DEQS score improves after eyewash, the possibility cannot be ruled out that there may be a bias that makes the untreated eye more uncomfortable. Thus, to provide more clues in relation to the tear function and ocular surface changes, we also carried out impression cytology analysis. Both MUC5AC and MUC16 expressions had no significant changes from baseline to one month after initiation of eyewash use. In the ratios of the areas stained with MUC16 antibodies to the areas of conjunctival epithelial cells, PAS staining of the conjunctival imprints, and MUC5AC plus goblet cells, there were no significant differences between before and after eyewash. Previously, we reported that the MUC5AC mRNA expressions and the goblet cell densities showed a drastic reduction in atopic patients with severe epithelial disease [17]. Furthermore, plenty of positive PAS stains for mucins in impression cytology specimens for eyes with significant epithelial disease and corneal ulcers were found. In contrast, non-goblet cell mucins (e.g., MUC1, 2, 4, and 16) mRNA expressions were all significantly upregulated in eyes with significant epithelial disease as compared with healthy control eyes [8]. Therefore, the ocular surface epithelial cells might be secreting other mucins to compensate for the decrease of MUC5AC and to protect the ocular surface. For this reason, it is important to evaluate the presence or absence of corneal epithelial disorders before initiation of eyewash use, and we will investigate the mucins other than MUC5AC and MUC16 and the efficacy of eyewash use in patients with atopic dermatitis in the future.

In the treatment of seasonal allergic conjunctivitis (SAC) and perennial allergic conjunctivitis (PAC), topical antihistamines, mast cell stabilizers, nonsteroidal anti-inflammatory drugs, and steroids are usually employed, however, we sometimes encounter patients who are refractory to these medications. Although additional steroids may be required to relieve patient symptoms, they can cause adverse effects such as cataract, infection, steroid-induced elevation of intraocular pressure (IOP), glaucoma, eyelid skin atrophy, and depigmentation with barrier impairments [18,19,20]. To overcome these problems, additional or alternative therapeutic medications have been required. Recently, eyewash has been used in patients with SAC or PAC. In the past, we reported that the antipruritic effect lasted significantly up to 20 min after washing the eyes [2]. We believe that eyewash treatment is mainly effective for early-phase reaction driven by acute histamine release in that it contains an antihistamine, chlorpheniramine maleate. Moreover, the effect of eyewash on acute allergic symptoms is also expected to be due to mechanical removal of the allergen from the ocular surface by washing, however, the patients tend to rub their eyes after this. Eyelid skin pollinosis (pollen dermatitis or pollen blepharitis) has been reported suggesting that adhesion of pollens to the thin and soft eyelid skin easily induces skin allergy [21,22]. Since itching occurs in the eyelid skin rather than the eyeball, ocular instillation may not be as effective as eyewash use in such patients. Treatment of the eyelid may be required for these patients [23,24] and eyewash solutions can be one of the methods for treating or preventing itching without any adverse drug reactions. Although this was a one month study, eyewash may be used for a longer period of time in the treatment of PAC. In addition, contact lens wearers were excluded in this study, but we should also study the efficacy and safety of eyewash in contact lens wearers. Therefore, we should consider a more extensive future study on this topic. 

The results of this study suggested that one month use of a nonpreserved eyewash solution has no adverse effects on the tear film and the ocular surface mucins.

## 4. Materials and Methods 

### 4.1. Study Design and Patient’s Information

This study adhered to the tenets of the Declaration of Helsinki. All procedures were performed in compliance with the protocol (#610) approved by the institutional ethics review board of the Tsurumi University School of Dental Medicine. Written informed consent was obtained from all patients.

We performed a prospective, contralaterally controlled study at the outpatient clinic in the Department of Ophthalmology, Tsurumi University Hospital. A total of 42 eyes in 21 individuals (17 males, 4 females; mean age: 36.1 ± 7.4 years) without ocular allergies, dry eyes, or other ocular diseases were analyzed. Eyes were randomized to a wash group (group one, *n* = 21) and a nonwash follow-up group (group two, *n* = 21). The following evaluation items at the baseline and after 1 month were compared to analyze within-subgroup variation and between-subgroup variation; DEQS, tear film BUT, fluorescein staining score, mRNA expression of MUC5AC and MUC16 in conjunctiva, MUC16 immunohistochemistry for the ratio of stained areas to the overall area of cellular pick-up, and MUC5AC PAS staining for goblet cell density counts (Table 1).

Patients who had at least one of the following 11 issues were excluded. Patients with: (1) a history of allergic hypersensitivity or known hypersensitivity to any compound or excipient of the Eyebon^®^, (2) any ocular condition that could affect their safety or study parameters (narrow angle glaucoma requiring medication or laser treatment, clinically significant blepharitis, follicular conjunctivitis, iritis, pterygium, or a diagnosis of dry eye), (3) a history of vernal keratoconjunctivitis or atopic keratoconjunctivitis, (4) ocular surgical intervention the past 3 months prior or refractive surgery in the past 6 months prior to study, (5) a presence of active ocular infection (bacterial, viral, or fungal) or preauricular lymphadenopathy, or an ocular herpetic infection, (6) any uncontrolled systemic disease, (7) a history of status asthmaticus, persistent moderate or severe asthma, or moderate to severe allergic asthmatic reactions to study allergens, (8) manifested signs or symptoms of clinically active allergic conjunctivitis (defined as the presence of any itching or >1 hyperemia in the conjunctival vessel bed) in either eye on their first visit. Patients who (9) are currently pregnant, nursing, or planning a pregnancy, (10) wear contact lenses, (11) have used any medications (topical, topical ophthalmic, systemic or injectable) during the period indicated before and during the study period. These included aspirin, aspirin containing products, H1-antagonist antihistamines (including ocular), and all other anti-allergy therapies, which include prescriptions, over-the-counter, homeopathy agents, all other topical ophthalmic preparations (including tear substitutes) other than study drops, corticosteroids or mast cell stabilizers, and depot corticosteroids.

### 4.2. Eyebon^®^

Eyebon^®^ is a commercial eyewash solution used in animal experiments and human clinical examinations, which can be used, up to six times/day, on the basis of the manufacturing and marketing approval standards for over-the-counter (OTC) drugs. The wash should occur for ≤30 s per administration based on a report on corneal epithelium disorders [1]. Firstly, cosmetics and blots around the eyes were wiped off before use. Secondly, the solution was poured, up to 5 mL, into the attached eyewash cup, and pressed firmly to the eye. Finally, while pressing the cup to the eye, the patient faced up by laying head back taking care not to spill the solution, and the eye was washed with several blinks. There are many kinds of Eyebon^®^; we used “Eyebon^®^ W vitamin”. The solution does not contain BAK, but the following active ingredients (in 100 mL): Dipotassium glycyrrhizinate 25 mg, chlorpheniramine maleate 3 mg, taurine 100 mg, pyridoxine hydrochloride (vitamin B6) 10 mg, cyanocobalamin (vitamin B12) 1 mg, and chondroitin sulfate sodium 10 mg. The following excipients are also included (in 100 mL): boric acid, borax, polysorbate 80, sodium edetate, propylene glycol, l-menthol, dl-camphor, and pH regulator.

### 4.3. Questionnaire

The DEQS score is a validated Japanese dry eye-specific questionnaire that has been used to assess the symptoms and their effects on the quality of life (QOL) in Japanese individuals [25]. The DEQS consists of 15 items related to dry eye symptoms and influence on daily life, and the overall degree of QOL impairment is calculated as a summary score from 0–100 points.

### 4.4. Tear Function Tests and Ocular Surface Staining 

We assessed the TBUT and the corneal and conjunctival fluorescein staining scores before and after the study, as reported previously [26,27]. The TBUT score was measured with observations of the cornea and conjunctiva through a slit lamp after instillation of a 2 μL of fluorescein solution into the conjunctival sac. The time from a normal blink to the first appearance of a dry spot in the tear film was measured. The fluorescein ocular surface staining was performed with a blue-free barrier filter. The ocular surface was divided into the following three zones: the nasal bulbar conjunctiva, the cornea, and the temporal bulbar conjunctiva. The maximum staining score for each area was 3 points, and the maximum staining score for the overall surface was 9 points. 

### 4.5. Conjunctival Impression Cytology

The impression cytology specimens were obtained after administration of topical anesthesia with 0.4% oxybuprocaine. Three separate strips of cellulose acetate filter paper (Millipore HAWP 304, Bedford, MA, USA) that had been soaked in distilled water for few hours and dried at room temperature (25 °C) were applied on the superior temporal, inferior temporal, and inferior nasal bulbar conjunctiva. These areas were pressed gently by a glass rod and then removed (Figure 8). The specimens were then fixed with formaldehyde. 

The specimens from superior temporal bulbar conjunctiva were for PCR of MUC5AC and MUC16. The specimens from inferior temporal bulbar conjunctiva were allocated to immunohistochemical staining to confirm the presence of MUC16. The specimens from inferior nasal bulbar conjunctiva were stained with PAS for MUC5AC, dehydrated in ascending grades of ethanol and then with xylol, and finally cover slipped. 

### 4.6. Quantitative Real-Time PCR for MUC5AC and MUC16 mRNA Expression

RNA was extracted from the isogen samples. A qRT-PCR was performed according to the manufacturer’s instructions (Applied Biosystems, Weiterstadt, Germany). The cDNA (10 ng) was amplified in 30 μL final volume in the presence of 1.2 μL of the ”Assay by Design” oligonucleotides (MUC5AC and MUC16 and GAPDH; Applied Biosystems). Test gene primer and probe sets were optimized for concentration, amplification efficiency, and faithful co-amplification with a housekeeper gene primer and probe sets, the latter including GAPDH. A qRT-PCR was set up in 96-well plates using the above reagents and TaqMan master mix and as indicated by optimization data and it was run on 7000 ABI thermal cyclers (Applied Biosystems). The thermal profile consisted of 50 °C for 2 min, 95 °C for 2 min, followed by 40 cycles of 95 °C for 15 sec and 60 °C 30 s. Real-time data were acquired and analyzed using Sequence Detection System Software (Applied Biosystems) with manual adjustment of the baseline and threshold parameters. The expression levels of mRNA were normalized by the median expression of a housekeeping gene (GAPDH). Real-time PCR primer sequences are shown in Table 2.

### 4.7. Immunohistochemistry Staining for MUC16

The impression cytology specimens for MUC16 immunohistochemistry staining were placed in plastic fenestrated embedding cassettes (Murazumi, Osaka, Japan). Then they were immersed in glass jars containing 10 mM of sodium citrate buffer (pH 6.0) and were subjected to microwave treatment at 500 W for 5 min to activate the antigen. The specimens were then washed in phosphate-buffered saline (PBS) twice for 5 min. The specimens were placed on slides and blocked in 5 mL PBS with 2 drops horse serum albumin for 20 min. The primary mouse monoclonal MUC16 antibodies at a dilution of 1:50 (PROGEN Biotechnik GmbH, Heidelberg, Germany) were applied for 24 h at room temperature (25 °C) in a moist chamber and then the slides were rinsed with PBS 3 times for 5 min. Samples were processed using the VECTASTAIN ABC Kit (DAKO, Co, Carpinteria, CA, USA) protocol (3,3′ diaminobenzidine-peroxide staining). In brief, the specimens were developed with 0.025% 3,3′ diaminobenzidine (Sigma Chemical, Co, St. Louis, MO, USA) mixed with 0.3% H_2_O_2_, and counterstained with hematoxylin. Dehydration by immersion in successively more concentrated solutions of ethyl alcohol, followed by immersion in xylene overnight. Finally, the specimens were transferred to glass bottles and cover slipped with mounting medium for light microscopic examination. The evaluation of specimens under light microscopy for the presence of positive immunohistochemical staining was also performed in a masked fashion.

### 4.8. MUC5AC Periodic Acid Schiff (PAS) Staining for Goblet Cell Density Counts

The quantitative studies of conjunctival goblet cells were conducted by taking photographs using a calibrated grid under a light microscope at a magnification of 400×. We photographed three nonoverlapping areas of each sample selected at random and averaged the outcomes for a single sample score. The goblet cell density counts were reported as cells per high power field (GC/HPF). The same researcher who was matched to whom the samples came from evaluated the specimens for goblet cell counts and mucin pick up.

### 4.9. Statistical Analysis

The data was analyzed using Prism software (ver. 8.1.0 for Mac; GraphPad Software Inc, San Diego, CA, USA) and Excel for Mac (ver. 16.16.11; Microsoft Corporation, Redmond, WA, USA). To compare differences in clinical ocular parameters between the groups, a paired t-test was used. All data was expressed as the mean ± standard deviation (SD). A *p*-value < 0.05 was considered to indicate statistical significance.

## Figures and Tables

**Figure 1 ijms-20-05078-f001:**
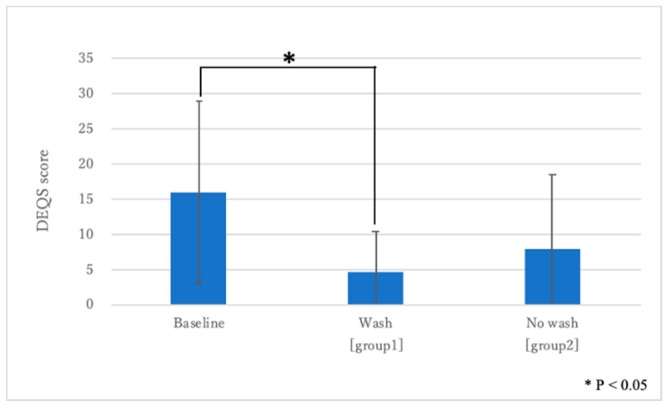
Changes in the dry eye-related quality-of-life score (DEQS) score. Note the significant improvement between before and after 1 month of eyewash use (*p* = 0.001).

**Figure 2 ijms-20-05078-f002:**
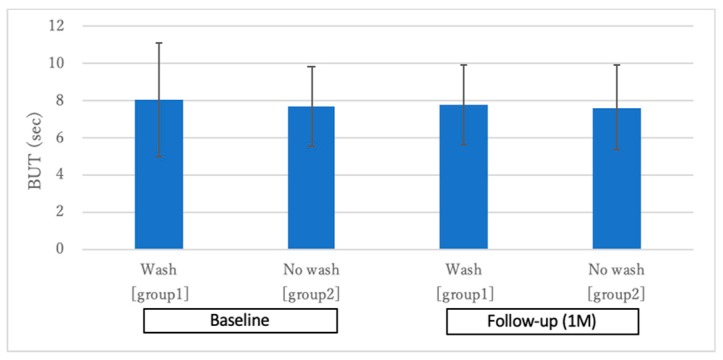
Changes in tear film breakup time (TBUT). Note the nonsignificant TBUT changes between before and after 1 month of eyewash use (*p* > 0.05).

**Figure 3 ijms-20-05078-f003:**
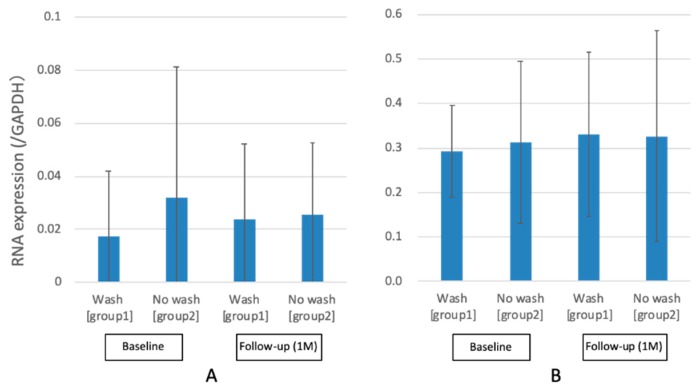
Changes in MUC5AC and MUC16 mRNA expressions. Note the nonsignificant MUC5AC and MUC16 RNA expression level changes between before and after 1 month of eyewash use (*p* > 0.05): (**A**) MUC5AC and (**B**) MUC16.

**Figure 4 ijms-20-05078-f004:**
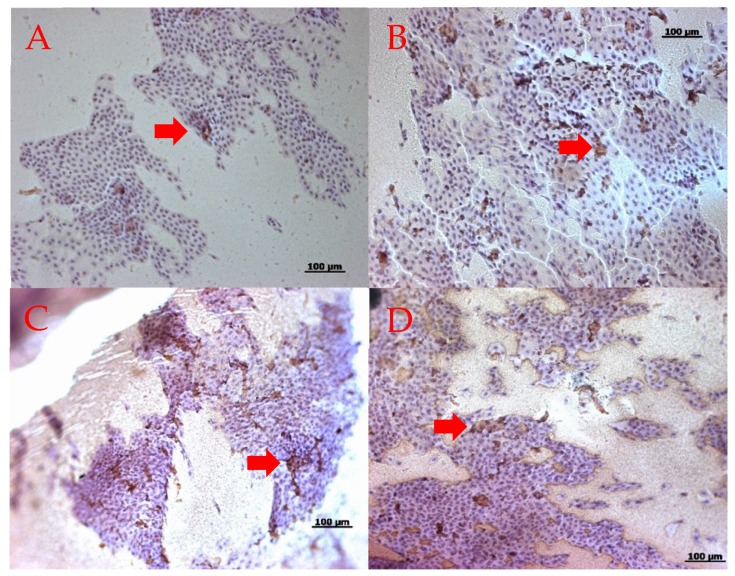
Representative MUC16 immunohistochemistry staining. Note the positive MUC16 staining before and after 1 month of eyewash use (red arrows): (**A**) baseline in group one; (**B**) after 1 month in group one; (**C**) baseline in group two; and (**D**) after 1 month in group two.

**Figure 5 ijms-20-05078-f005:**
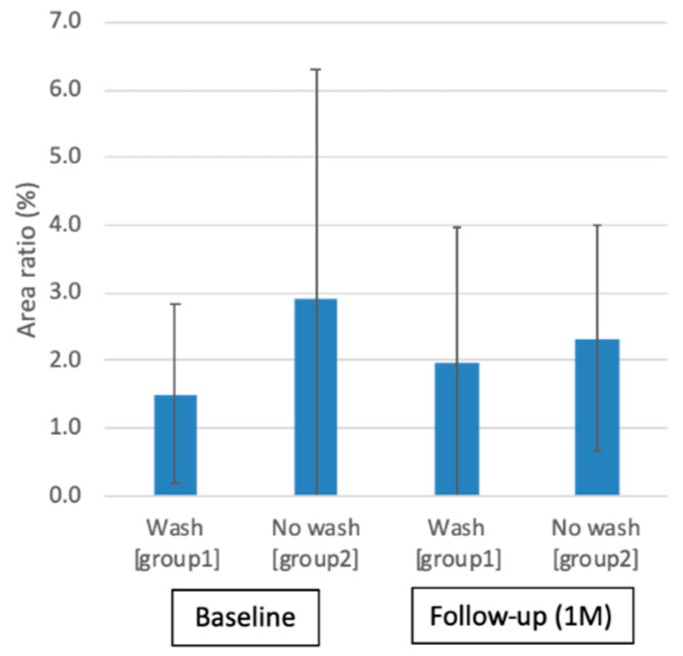
Ratio of areas stained with MUC16 antibodies to overall area of cellular pick-up. Note the nonsignificant MUC16 antibody staining changes between before and after 1 month of eyewash use (*p* > 0.05).

**Figure 6 ijms-20-05078-f006:**
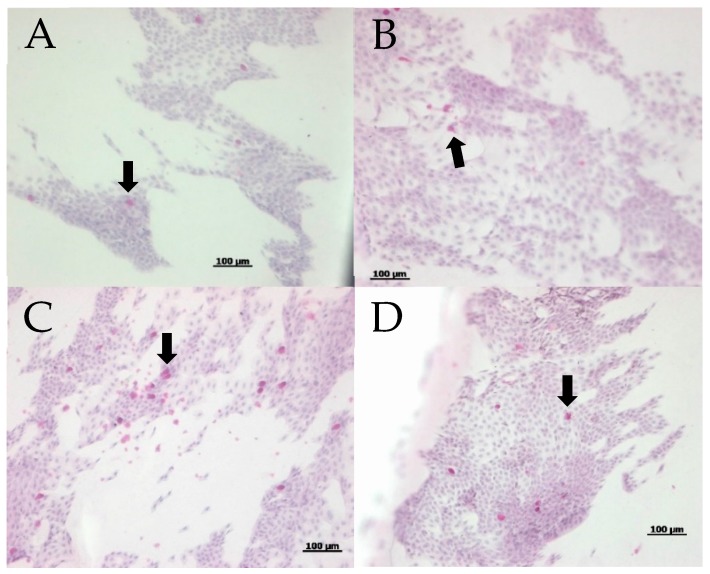
Representative MUC5AC PAS staining. Note the numerous MUC5AC + goblet cells before and after 1 month of eyewash (black arrows): (**A**) baseline in group one; (**B**) after 1 month in group one; (**C**) baseline in group two; and (**D**) after 1 month in group two.

**Figure 7 ijms-20-05078-f007:**
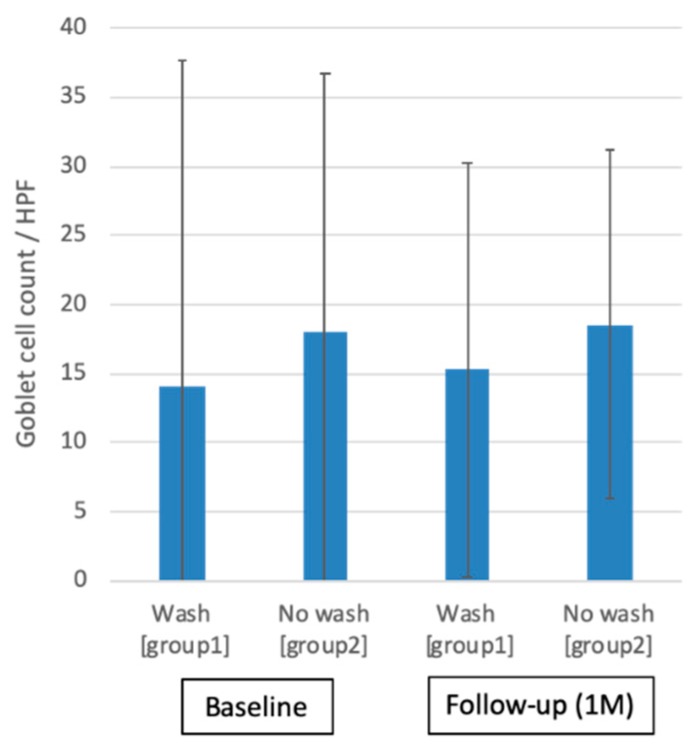
Goblet cell count in periodic acid Schiff (PAS) staining. Note the nonsignificant changes in goblet cell count between before and after 1 month of eyewash use (*p* > 0.05).

**Figure 8 ijms-20-05078-f008:**
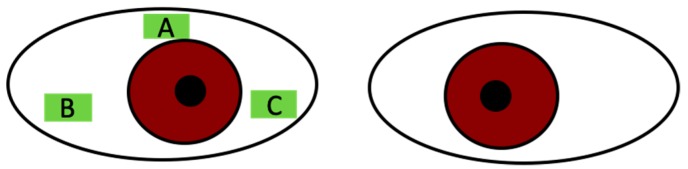
Areas of impression cytology: (**A**) superior bulbar conjunctiva for RT-PCR of MUC5AC and MUC16, (**B**) inferior temporal bulbar conjunctiva for MUC16 immunohistochemistry, and (**C**) inferior nasal bulbar conjunctiva for MUC5AC PAS stain.

**Table 1 ijms-20-05078-t001:** Evaluation items at baseline and after 1 month of eyewash use.

	Baseline		1 Month	
	Group 1	Group 2	Group 1	Group 2
DEQS	○		○	○
BUT (sec)	○	○	○	○
MUC5AC (mRNA)	○	○	○	○
MUC16 (mRNA)	○	○	○	○
MUC16 immunohistochemistry stain	○	○	○	○
MUC5AC PAS stain	○	○	○	○

○: conducted

**Table 2 ijms-20-05078-t002:** Primers used for RT-qPCR analysis of genes.

	Primer Sequence (5′-3′)	
Gene	Forward	Reverse
*MUC5AC*	CAGGGCTGGTACACCTTGTC	ACGACATCTGCATCGATTGGA
*MUC16*	GCCTCTACCTTAACGGTTACAATGAA	GGTACCCCATGGCTGTTGTG

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
