# Peer review of "The Effect of Long-Term Use of an Eyewash Solution on the Ocular Surface Mucin Layer"

_ijms, 2019, doi:10.3390/ijms20205078_

Round 1

Reviewer 1 Report

In this paper, authors have undertaken a series of experiments and shown that the use of non-preserved eyewash solution has no detrimental effects on the tear film and the ocular surface mucins. The results of this study give the clear evidence to ophthalmologists' concern that washing eyes may affect the structure of the mucin and promote infection and epithelial keratoconjunctivits. However, there are some concerns to be considered below.

In Figure 1, DEQS score at baseline should show as group 1 and group 2, not as total number. Then, the statistical analysis should perform between baseline and 1 month for group 1 and group 2.

This study uses healthy subjects without ocular allergies, dry eyes, or other ocular diseases. Is there any concern if such patients use non-preserved eyewash solution? If there is any concern, authors should add some discussion in the manuscript.

Authors focus the safety of non-preserved eyewash solution in this paper. Because the reader of this journal is not limited to the ophthalmologist, describe some reason or merit why the patients with allergic conjunctivitis use the eyewash solution.

Author Response

We thank the editors and the reviewer1 for the interest in our manuscript and the constructive comments. We have given careful consideration to all comments, and are grateful for the opportunity to improve our manuscript.

â‘ In Figure 1, DEQS score at baseline should show as group 1 and group 2, not as total number. Then, the statistical analysis should perform between baseline and 1 month for group 1 and group 2.

→Thank you for your great comment. We revised sentence as follows; 

Page2, Line61-62 ;...(P = 0.001), whereas in group 2, there was no significant difference between baseline and 1 month later. (P > 0.05)...

â‘¡This study uses healthy subjects without ocular allergies, dry eyes, or other ocular diseases. Is there any concern if such patients use non-preserved eyewash solution? If there is any concern, authors should add some discussion in the manuscript.

→Thank you for your excellent suggestion. We think The effect of the use of eyewash solutions in eyes with such diseases and low mucin expression levels should be investigated soon. If the microenvironment of the ocular surface in such patients is different from that in healthy people, it cannot be denied that mucin is decreased. Therefore, in the present study, we intend to study only healthy people, then patients with any ocular diseases in the future. However, there are ethical problems in Japan, so it is a future issue. So we added the sentence as follows;

Page6, Line173-177 ; …If the microenvironment of the ocular surface in such patients is different from that in healthy people, it cannot be denied that mucin is decreased. Therefore, in the present study, we intend to study only healthy people, then patients with any ocular diseases in the future. However, there are ethical considerations in Japan, so it is a future issue.

â‘¢Authors focus the safety of non-preserved eyewash solution in this paper. Because the reader of this journal is not limited to the ophthalmologist, describe some reason or merit why the patients with allergic conjunctivitis use the eyewash solution.

→Thank you for your fabulous comment. We mentioned about merit in discussion section (Page7, Line198~). But we added sentence as follows;

Page7, Line212-216 ; ...after washing the eyes[2]. We believe that eyewash treatment is mainly effective for early-phase reaction driven by acute histamine release in that it contains an antihistamine, chlorpheniramine maleate. Moreover, effect of eyewash on acute allergic symptoms is also expected to be due to mechanical removal of the allergen from the ocular surface by washing. However....

Reviewer 2 Report

The experimental design is well organized, the results are shown in clear manner and the discussion are well structured. Nevertheless, I believe that there are some points that the authors should clarify:

- the volunteers undergo one eye to wash and the other no, could this way of treatment cause an incorrect perception of dry eye-related quality-of-life by individuals?

- the formulation Eyebon® contains boric acid and borax so, what is the pH regulator? Furthermore, among the ingredients there are some possible products with irritant activity as polysorbate 80, l-menthol and dl-camphor. Did the authors evaluate this?

- at the end of the discussion section the authors report the use of the eyewash in contact lens wearers but this statement is speculative since no studies have been carried out in this sense and authors themselves state that they must be performed.

Author Response

We thank the editors and the reviewer2 for the interest in our manuscript and the constructive comments. We have given careful consideration to all comments, and are grateful for the opportunity to improve our manuscript.

â‘ The volunteers undergo one eye to wash and the other no, could this way of treatment cause an incorrect perception of dry eye-related quality-of-life by individuals?

→Thank you for your great comment. Certainly, the possibility of bias is undeniable.We revised sentence as follows;

Page6, Line186-188 ;...However, if DEQS score improve after eyewash, the possibility cannot be ruled out that there may be a bias that makes the untreated eye more uncomfortable...

â‘¡The formulation Eyebon® contains boric acid and borax so, what is the pH regulator? Furthermore, among the ingredients there are some possible products with irritant activity as polysorbate 80, l-menthol and dl-camphor. Did the authors evaluate this?

→Thank you for your excellent comment. Eyebon is added the pH regulator because it could be adjusted to the same pH as the tear fluid to prevent it from burning sensation. As you pointed out, we cannot deny the possibility that excipients such as polysorbate 80 also may have activity. However, we did not evaluate it this time, so we added about that as follows;

Page6, Line177-181 ;...Eyebon® is added the pH regulator because it could be adjusted to the same pH as the tear to prevent it from burning sensation. We cannot deny the possibility that excipients such as polysorbate 80, l-menthol and dl-camphor also may have activity of epithelial damage like BAK, but we did not evaluate it this study. Thus, further study to examine whether corneal epithelial disorders can be caused by excipients must be needed.

â‘¢At the end of the discussion section the authors report the use of the eyewash in contact lens wearers but this statement is speculative since no studies have been carried out in this sense and authors themselves state that they must be performed.

→Thank you for your fabulous comment. As you pointed out, contact lens is excluded in this study, so it is just a guess. However, we are currently studying the efficacy and safety of eyewash in contact lens wearers and will report this in the near future. So we revised sentence as follows;

Page7, Line222-225 ; in the treatment of PAC. In addition, contact lens wearers were excluded in this study,but we should also study the efficacy and safety of eyewash in contact lens wearers. Therefore, we should consider a more extensive future study on this topics.